# Surface Quality Control Strategy of Aspherical Mold Based on Screw Feed Polishing of Small Polishing Tool

**DOI:** 10.3390/ma15144848

**Published:** 2022-07-12

**Authors:** Jiarong Zhang, Han Wang, Xiangyou Zhu, Honghui Yao, Shaomu Zhuo, Shuaijie Ma, Daohua Zhan, Nian Cai

**Affiliations:** 1State Key Laboratory of Precision Electronic Manufacturing Technology and Equipment, Guangdong University of Technology, Guangzhou 510006, China; wanghangood@126.com (H.W.); 13631436831@163.com (X.Z.); gdut_m@163.com (S.Z.); shuaijiem817@163.com (S.M.); zhandaohua@mail2.gdut.edu.cn (D.Z.); hi_super_zhang@163.com (N.C.); 2Guangdong Provincial Key Laboratory of Micro-Nano Manufacturing Technology and Equipment, Guangdong University of Technology, Guangzhou 510006, China

**Keywords:** small polishing tool, screw feed, surface error correction, surface roughness distribution, deterministic polishing

## Abstract

For small aspherical molds, it is difficult for the existing polishing method to take into account the correction of the surface error and the control of the uniformity of the surface roughness (SR) distribution, because the polishing tool is always larger than the small mold. Therefore, we used viscoelastic polyester fiber cloth to wrap the small steel ball as a polishing tool to adapt to the surface shape change of the aspherical mold, and designed a semi-flexible small polishing disc tool with microstructure, which can better adapt to the curvature change of aspherical surface and obtain better SR Ra. At the same time, a combined polishing method of constant speed and variable speed for screw feed was proposed to improve the uniformity of SR distribution in the paper. Then, a series of theoretical analysis and experimental verification were carried out in this paper to predict the tool influence function (TIF) of the two polishing tools and the effectiveness of the combined polishing method. In the experiment, a TIF bandwidth of about 0.46 mm was obtained with a small spherical polishing tool, which favors the surface shape correction of the small aspherical mold. The experiment of uniform removal with a small polishing disc tool was carried out to quickly reduce the Ra. Finally, the surface quality of the aspherical mold was effectively improved, combined with the constant speed and variable speed polishing modes of screw feed of the small spherical polishing tool and the smoothing effect of the small polishing disc tool. The peak valley (PV) of two small aspherical molds with an optical effective diameter less than 13 mm converged from 0.3572 μm and 0.2075 μm to 0.1282 μm and 0.071 μm, respectively. At the same time, the SR dispersion coefficient was reduced from 27.9% and 41.6% to 14.2% and 12.7%, respectively. The study provides a good solution for the surface quality control of small aspherical molds.

## 1. Introduction

Aspherical lenses are effective in simplifying the design of optical systems. They can also correct the aberrations, improve the optical performance, reduce the size and weight of the optical system, and lower the cost. They are widely used in modern optical systems [1,2].

At present, small glass aspherical optics are mainly manufactured in batches through a hot molding process [3]. In addition, the processing quality of optical molds is one of the most intuitive factors affecting the performance of glass aspherical lenses. Therefore, the machining precision must be higher than the product accuracy after molding [4]. Mainstream precision polishing methods for small aspherical molds include jet polishing, ion beam polishing, magnetorheological polishing, airbag polishing, and contact polishing with small polishing tools, etc. Jet polishing relies on the jet shock of polishing grains to remove materials, and is suitable for polishing small deep concave aspherical molds. However, the polishing removal rate is mediocre, while the TIF is unstable [5]. By contrast, ion beam polishing removes materials through the physical sputtering effect of a specific ion beam bombardment on the optical workpiece surface. It will avoid contact with the workpieces, making it easy to obtain an undamaged surface. Moreover, it can form a small TIF by designing an ultra-small caliber fence to polish and shape the mold with high precision. However, this polishing method has high requirements for the processing environment while the process is complicated and the material removal efficiency turns out to be low [6]. Therefore, this method is more suitable for high-precision polishing of large-diameter workpieces with a high price rather than for the polishing of small-caliber aspherical molds which require high efficiency. With the magnetorheological polishing method, the machining process is controllable, and the results are accurate and measurable. This method can achieve better surface shape accuracy and lower SR than conventional machining, and produce the surface with almost no damage. However, conventional magnetorheological polishing employs a magnetic wheel as a polishing tool, which is hard to adapt to a small concave aspherical workpiece [7]. Maloney et al. [8] of QED Company developed a magnetorheological micro-polishing wheel with a diameter of 10 mm, which still cannot polish a concave aspherical surface with a curvature radius of less than 7 mm. Yin et al. [9] developed an oblique-axis magnetorheological polishing technology with a small cylindrical permanent magnet as the polishing tool. The SR was reduced to Ra 1 nm after polishing the concave aspherical mold with a diameter of 8 mm. However, the total polishing time was as long as 120 min, and the surface repair work of the mold was not carried out. Recently, we [10] designed a spherical permanent magnet spherical polishing tool with a diameter of 2.4 mm, and then performed magnetorheological polishing experiments on an aspherical mold with a diameter of 11.2 mm on the self-developed polishing equipment. The SR and surface shape accuracy decreased from Ra 8.2121 nm to Ra 1.6745 nm and PV 193.0 nm to PV 105.4 nm, respectively. However, there is still the disadvantage of a long polishing time. Furthermore, it is more difficult for traditional disc, wheel-shaped, or airbag polishing tools to adapt to the change of curvature at each point of the small-diameter non-spherical optical elements. Even worse, such small polishing tools are more prone to wear and need to be shaped constantly in order to ensure the accuracy of processing, which seriously affects the processing quality and efficiency [11,12]. As a result, with the traditional contact-type polishing methods, it is hard to simultaneously take into account the correction efficiency of both SR and surface PV (especially of deep concave shapes).

On the other hand, the high-frequency error (HFE) also has a great influence on the optical performance. It is particularly important to reduce the HFE while ensuring the low-frequency surface accuracy during the polishing. For example, in our previous work, we proposed a combined process of small ball-end contact polishing and magnetorheological polishing for small aspherical molds, which can reduce the SR from Ra 8.5972 nm to Ra 1.2694 nm [10]. H. M. Martin et al. [13] used stress disk polishing technology to suppress mid-high frequency errors, and polished the primary mirror of an 8.4 m LBT telescope to a surface accuracy of RMS 18 nm. DeGroote Nelson et al. [14] used a high-frequency vibration method with full aperture coverage, which can also suppress HFE well. Dunn C. R. et al. [15] have done research on polishing trajectory and have proven that pseudo-random polishing trajectory can significantly reduce the mid-high frequency error. The above studies focus on HFE improvement methods for workpieces, and have achieved effective results. But this research ignores the phenomenon of uneven HFE distribution on the surface of the workpiece after polishing due to differences in polishing methods or polishing trajectories, causing the optical element to produce different angles of scattering at different positions, which seriously affects the optical performance [16].

In addition, the screw feed method is usually used to polish rotary symmetrical workpieces. In order to ensure a consistent material removal rate, some scholars [16,17] have paid attention to the overlap rate to achieve the same overlap area of removed areas in spiral trajectory polishing, while few of them have explored the influence of the screw feed polishing methods of different rotation modes of workpiece axes on the surface quality and efficiency.

On the basis of the above discussion, the authors of this paper design a small spherical polishing tool wrapped with viscoelastic polyester fiber cloth and a semi-flexible disc-shaped polishing tool with microstructure, and establish the TIF of the two polishing tools. Then, combined with the workpiece axis constant speed polishing mode and variable speed polishing mode, the authors solve the problems of difficult surface PV repair and poor SR and SR distribution uniformity of small-diameter aspherical molds. Finally, this paper provides a good machining solution for surface quality control of small-diameter aspherical molds.

## 2. Material Removal Principle and Model for Small Polishing Tool

In polishing, the elastic small spherical polishing tool is in contact with the surface of the workpiece. It can be known from the Hertz contact theory [18] that the materials will deform locally to form an elliptical contact area when two surfaces are in contact and pressed tightly against each other. It is much smaller than any radius of curvature in the aspherical surface, so it can be assumed that the contact region is approximately circular. In addition, J.A. Greenwood [19] believes that the difference of contact pressure is less than 3% when the elliptic contact area is converted to a circular area, so it is reasonable to convert them for most cases. We make the following assumptions in the subsequent modeling to further simplify the solution of the TIF:(1)The polishing tool wrapped with flannel is regarded as a standard ball tool.(2)The polishing tool and workpiece are considered as rigid bodies.(3)The polishing particles uniformly adhere to the surface of the polishing tool.

According to the Preston equation, the TIF obtained after the integral normalization can be expressed as [20]:(1)$$R\left(x,y\right)=\frac{k_{p}}{T}\int_{0}^{T}p\left(x,y\right)v\left(x,y\right)dt$$

In Formula (1), R(x,y) refers to the unit removal function, $k_{p}$ stands for the removal coefficient, T is the unit time, p(x,y) means the pressure distribution function of the contact surface, v(x,y) stands for the relative line speed distribution function of the contact surface, and dt suggests the residence time of the small polishing tool on the workpiece surface.

### 2.1. Modeling of Pressure Distribution of Small Polishing Tool Contact Area

#### 2.1.1. Modeling of Pressure Distribution of Small Spherical Polishing Tool

The pressure distribution in the contact region can be expressed as Equation (2) according to the Hertzian contact theory [21]:(2)$$P\left(x,y\right)=P_{0}\sqrt{1-\frac{x^{2}+y^{2}}{a^{2}}}.$$

The contact circle radius of the small spherical polishing tool is $a=m{\left[\frac{3Fr_{e}}{4E^{*}}\right]}^{\frac{1}{3}}$ and the central point pressure is $P_{0}=\frac{3F}{2\pi a^{2}}$, wherein $E_{1}$ and $E_{2}$ are the elastic moduls and $\varepsilon_{1}$ and $\varepsilon_{2}$ are the Poisson ratios of the polishing tool and workpiece. In addition, m is the contact radius correction coefficient, while F stands for the normal positive pressure during polishing. $r_{e}$, $r_{p}$, $r_{c}$, and $r_{m}$ are the equivalent curvature radius, the polishing tool curvature radius, and the maximum and minimum principal curvature radius of the contact region, respectively. Meanwhile, $r_{e}={\left[AB\left(\frac{A+B}{2}\right)\right]}^{\frac{1}{3}}$. In this equation, $\frac{1}{A}=\frac{1}{r_{p}}+\frac{1}{r_{c}}$, $\frac{1}{B}=\frac{1}{r_{p}}+\frac{1}{r_{m}},\;\mathrm{and}\;E^{*}=\frac{E_{1}E_{2}}{\left(1-\varepsilon_{1}^{2}\right)E_{2}+\left(1-\varepsilon_{2}^{2}\right)E_{1}}$.

#### 2.1.2. Modeling of Pressure Distribution of Small Polishing Disc Tool

The bottom surface of the small polishing disc tool is a flexible plane of small size. Thus, it can be assumed that the contact circle radius is equal to the polishing tool radius, and the pressure in the contact area can be approximately expressed as:(3)$$P{\left(x,y\right)}^{\prime }=\frac{F}{\pi r_{p}{}^{2}}.$$

### 2.2. Modeling of Speed Distribution of Small Polishing Tool Contact Area

#### 2.2.1. Modeling of Speed Distribution of Small Spherical Polishing Tool

(1) Workpiece Axis Constant Rotation Speed Mode

As shown in Figure 1a, the global coordinate system *O-XYZ* is established.

First, the small polishing tool is fed along a generatrix of the aspherical surface in polishing before establishing the coordinate system $o_{p}x_{p}y_{p}z_{p}$ in the center of the small spherical polishing tool $o_{p}$. Then, the polishing tool rotates the axis $z_{p}$ around with the angular velocity $\omega_{p}$ and keeps the direction of $\omega_{p}$ the same as that of axis $z_{p}$. Thereinto, the angle of the axis $z_{p}$ rotating around the *x*-axis is σ, so the polishing axial angular velocity vector $\vec{\omega_{p}}$ can be expressed as:(4)$$\vec{\omega_{p}}=\omega_{p}{\vec{z}}_{p}=\omega_{p}\left(0,\sin\sigma y,\cos\sigma z\right).$$

$\vec{\;r\;}$ is the vector from the spherical center $o_{p}=\left(0,0,r_{p}\right)$ to point *A*, and $r_{p}$ is the radius of the spherical polishing tool, forming a formula as:(5)$$\vec{\;r\;}=\vec{o_{p}A}=\left(x_{a},y_{a},-r_{p}\right).$$

Therefore, the line speed $\vec{v_{p}}$ generated by the small spherical polishing tool in the polishing area can be expressed as:(6)$$\vec{v_{p}}=\vec{\omega_{p}}\times \vec{\;r\;}=\omega_{p}\left(-\sin\sigma r_{p}-\cos\sigma y_{a},\cos\sigma x_{a},-\sin\sigma x_{a}\right).$$

Meanwhile, the workpiece axis rotates around axis $Z_{c}$ at the angular speed of $\omega_{c}$, and the center point of the aspherical mold *O* is at the rotation center of the workpiece axis; the angle between rotor $Z_{c}$ and axis *Z* is the aspherical normal rake β.

Similarly, the line speed $v_{c}$ generated by the workpiece axis at point *A* can be expressed as:(7)$$\vec{v_{c}}=\vec{\omega_{c}}\times \vec{\;OA\;}=\left(\sin\beta \omega_{c},0,\cos\beta \omega_{c})\times (r+x_{a},y_{a},Z+z_{a}\right)\;=(-\cos\beta \omega_{c}y_{a},\cos\beta \omega_{c}\left(r+x_{a}\right)-\sin\beta \omega_{c}\left(Z+z_{a}\right),\sin\beta \omega_{c}y_{a}).$$

The feed velocity along the diameter of the workpiece is so small that it can be considered negligible. In addition, the curvature of the contact area is also small, so the paper provides that the contact circle is in the *oxy* plane and $z_{a}\;$= 0. The relative line speed $v_{a}$ of point *A* is composed of the line speed $v_{c}$ produced by the workpiece rotation and the line speed $\;v_{p}$ produced by the rotation of the spherical polishing tool, as shown in Figure 1b. Finally, the following equation is formed:(8)$$\vec{v_{a}}=\vec{v_{p}}-\vec{v_{c}}=\left[\begin{array}{c} \;v_{ax}\\ v_{ay}\\ v_{az} \end{array}\right]=\left[\begin{array}{c} (-\sin\sigma r_{p}-\cos\sigma y_{a})\omega_{p}+\cos\beta \omega_{cy_{a}}\\ \cos\sigma \omega_{p}x_{a}-\cos\beta \omega_{c}\left(r+x_{a}\right)+\sin\beta \omega_{c}Z\\ -\sin\sigma x_{a}-\sin\beta \omega_{c}y_{a} \end{array}\right].$$

The TIF is only affected by the direction velocity of the tangent plane *oxy*, which is independent of the component velocity in the *z*-direction when polished with the small spherical polishing tool. Hence, the effective relative line velocity $v_{a}\left(x,y\right)$ acting on the TIF at point *A* can be obtained by:(9)$$v_{a}\left(x,y\right)=\sqrt{v_{ax}{}^{2}+v_{ay}{}^{2}}\;=\sqrt{({\left(-\sin\sigma r_{p}-\cos\sigma y_{a})\omega_{p}+\cos\beta \omega_{c}y_{a}\right)}^{2}+{\left(\cos\sigma \omega_{p}x_{a}-\cos\beta \omega_{c}\left(r+x_{a}\right)+\sin\beta \omega_{c}Z\right)}^{2}}.$$

(2) Workpiece Axis Variable Speed Mode

The paper takes the polishing residence site *o* as the analysis point, and makes $x_{a}=0$, $y_{a}=0$, so Formula (8) can be reduced to:(10)$${\left(\sin\sigma \omega_{p}r_{p}\right)}^{2}+{\left(-\cos\beta \omega_{c}r+\sin\beta \omega_{c}Z\right)}^{2}=v_{a}{}^{2}.$$

Therefore, the workpiece axis rotation speed $n_{c}$ of each polished residence point at the aspherical surface can be obtained:(11)$$n_{c}=\frac{30\sqrt{v_{a}{}^{2}-{\left(\sin\sigma \omega_{p}r_{p}\right)}^{2}}}{\pi \left|\begin{array}{c} \cos\beta r-\sin\beta Z \end{array}\right|}.$$

The paper takes the aspherical mold one in Figure 2a as an example, and sets $v_{a}=37\;\mathrm{mm}/s$, $n_{p}=200\;\mathrm{rpm}$, $\sigma =35^{{}^{\circ}}$ according to the polishing parameters of the constant rotation speed mode. Next, it obtains the relationship of the workpiece axis rotation speed with the aspherical mold radius according to Formula (11), as shown in Figure 2b. It can be seen that the rotational speed and the growth speed of the aspherical central area are relatively large. The paper sets the maximum rotational speed of the workpiece axis as $n_{c}=180\;\mathrm{rpm}$ after comprehensively considering the acceleration and maximum speed of the polishing equipment. Therefore, the variable speed mode explored in the study is composed of the constant line speed and the constant rotation speed of the contact region, as shown in Equation (12) and Figure 2b.
(12)$$n_{c}=\{\begin{array}{c} \;180\;,\;0\leq r<1.3\\ \;\frac{30\sqrt{v_{a}{}^{2}-{\left(\sin\sigma \omega_{p}r_{p}\right)}^{2}}}{\pi \left|\begin{array}{c} \cos\beta r-\sin\beta Z \end{array}\right|},\;r\geq 1.3 \end{array}.$$

#### 2.2.2. Modeling of Speed Distribution of Small Polishing Disc Tool

The polishing inclination *σ* is set to 0 when using a small polishing disc tool, as shown in Figure 3. The speed distribution of the contact area can be shown as Equation (13):(13)$$v_{a}^{\prime }\left(x,y\right)=\sqrt{v_{ax}^{\prime }{}^{2}+v_{ay}^{\prime }{}^{2}}=\sqrt{{\left(-\omega_{p}y_{a}+\cos\beta \omega_{c}y_{a}\right)}^{2}+{\left(\omega_{p}x_{a}-\cos\beta \omega_{c}\left(r+x_{a}\right)+\sin\beta \omega_{c}Z\right)}^{2}}$$

### 2.3. Residence Time Solution

The study employs the screw feed method as the polishing track and only requires a one-dimensional feed for axially symmetric aspherical rotation. Therefore, the TIF of the section profile loop band integral on the radial line of the aspherical mold can be converted from Equation (1) to Equation (14):(14)$$R\left(\rho ,\theta \right)=\frac{2k_{p}}{\omega_{c}}\int_{0}^{\theta_{0}}p\left(\rho ,\theta \right)v\left(\rho ,\theta \right)d\theta .$$
wherein, x=ρcos(θ),y=ρsin(θ). In this equation, ρ is the polar diameter and θ refers to the polar angle.

The small polishing tool is fed in a diametrical direction parallel to the *X-axis* of the polishing equipment during the polishing process of the rotary axial symmetric mold. It is assumed that the small polishing tool has 2*n* + 1 residence points in the workpiece diameter direction and 2*m* + 1 sampling points, in which *m* >> *n*.

Thus, the amount to be removed can be expressed as Equation (15):(15)$$\vec{H}={\left[H_{1},H_{2},\cdots H_{i},\cdots ,H_{m}\right]}^{T}.$$

The residency time vector is:(16)$$\vec{T}={\left[T_{1},T_{2},\cdots T_{j},\cdots ,T_{n}\right]}^{T}.$$

There are different TIFs at each sample point as shown in Figure 4, which can be represented by the discrete matrix of Equation (17):(17)$$\vec{R}=\left[R_{m1},R_{m2},\cdots R_{mk},\cdots ,R_{mn}\right],$$
where $\;R_{mk}$ is a vector composed of m sampling points along the diameter direction of the workpiece in the k-th resident point.

Taking the 2-norm of the formula $\vec{E}=\vec{R}*\vec{T}$ as the optimization objective and adopting the NNLS method, the corresponding residence time can be calculated as:(18)$$\min\|\vec{H}-\vec{R}*\vec{T}{\|}_{2}\;,T\geq 0.$$

### 2.4. Polishing TIF Simulation and Trajectory Planning

We carried out the simulation of the TIF of mold one combining with the parameters in Table 1, which are represented by discrete data points in concentric circles. Then, convolution iteration was performed on the removal of each sampling point to obtain the overall TIF, as shown in Figure 5a. However, this is time consuming, due to the frequent feeding and retracting operations required to polish the mold in the form of concentric circular trajectories. Therefore, we obtained the dwell time of every two adjacent dwell points on the mold by Formula (18), and performed screw feed polishing in the form of controlling the pitch. Figure 5b,c are the 2D and 3D images of the polishing trajectory of the equal pitch screw feed, respectively. It can be seen that the adjacent pitch of the 3D image varies with the curvature of the aspherical surface, and the helical line at the edge is relatively sparse. Furthermore, Figure 5d−f shows the corresponding simulation diagrams of the TIF and helical trajectory of the variable pitch. The spatial distribution of the pitch is more uniform, and the TIF of each resident point also tends to be uniformly removed after the convolution iteration.

## 3. Experiment Preparation

The study uses self-developed polishing equipment for experiments as shown in Figure 6. The aspherical mold contour formula can be stated as follows:(19)$$H=\frac{x^{2}}{R+\sqrt{R^{2}-(k+1)x^{2}}}+\overset{10}{\underset{j=1}{\sum }}A_{2j}x^{2j}.$$

In the formula, R is the radius of curvature of an aspherical vertex, k refers to the aspherical coefficient, and $A_{j}$ stands for the aspherical high-order term coefficient.

We prepared two tungsten carbide aspherical molds and one tungsten carbide plane mold for experiments as shown in Figure 2a, and measured the surface shape through the UA3P-300 surface profilometer as shown in Figure 7.

Then, we used viscoelastic polyester fiber cloth to wrap the small steel ball as a polishing tool to adapt to the surface shape change of the aspherical mold, as shown in Figure 8a. Furthermore, we designed a semi-flexible polishing tool to reduce the Ra, whose surface combined the microstructure of diamond particles, as shown in Figure 8b. The matching design of the flexible buffer layer and semi-rigid film can better adapt to the change of sphere curvature and obtain a better smoothing effect. The basic polishing and test conditions are shown in Table 1.

### 3.1. Verification of Spherical Small Polishing TIF

The removal depth of the contact center at the corresponding r-ring band position R(r) can be obtained by combining Formulas (1), (3) and (9), as below:(20)$$R\left(r\right)=k\frac{3F}{2\pi a^{2}}\sqrt{{\left(\sin\sigma \omega_{p}r_{p}\right)}^{2}+{\left(-\cos\beta \omega_{c}r+\sin\beta \omega_{c}Z\right)}^{2}}\cdot t.$$

The removal coefficient k can be obtained as:(21)$$k=\frac{R\left(r\right).2\pi a^{2}}{3F\sqrt{{\left(\sin\sigma \omega_{p}r_{p}\right)}^{2}+{\left(-\cos\beta \omega_{c}r+\sin\beta \omega_{c}Z\right)}^{2}}\cdot t}.$$

First, we designed six groups of single ring band polishing experiments with the workpiece axis speed as a single variable (Figure 9). The major experimental parameters and results are shown in Table 2, other polishing parameters (except the workpiece axis rotation speed) are shown in Table 1, while the ring band experimental contour measurement results are shown in Figure 10a.

The removal coefficient *k* obtained by the ring band experiment 1–4 shows an almost linear increasing trend in Table 2. It is not until a polishing time of about 226 s that the removal coefficient obtained in ring band experiment 4–6 tends to stabilize, which may be due to the fact that the polyurethane flannelette wrapped by the small polishing tool is insufficiently in contact with the free abrasive particles in the initial stage. Therefore, the number of effective polishing particles fails to reach the saturation state, explaining why the removal coefficient shows an increasing trend in the initial state. Thus, the relatively stable experimental results of ring band experiment 4–6 were selected to calculate the average of the removal coefficient, the bandwidth and the correction factor, $\bar{\;k}=5.792\times 10^{-6}$, $\bar{a}$ = 0.46 mm, and m=1.41, respectively. Then, the TIF verification experiments of ring bands 7 and 8 were carried out on the same tungsten carbide plane mold, where the workpiece axis speed was set as $v_{c7}=17\;\mathrm{mm}/s$ and $v_{c8}=23\;\mathrm{mm}/s$ while other polishing parameters remained consistent with that of ring band 1–6. The theoretical TIF matches with the actual removal profile shape at a high degree, as can be seen from Figure 10b.

### 3.2. Model Verification of Small Polishing Disc Tool

It was found that the small spherical polishing tool is easy to bring in a small-scale medium and HFE due to the small TIF, the unstable initial TIF, and the external factors, etc. Therefore, a small polishing disc tool structure with smoothing effect was designed to improve the HFE, as shown in Figure 8b. Then, a uniform polishing experiment was performed on the aspherical mold using the polishing parameters in Table 1 and Table 2 (Figure 11).

The results are shown in Figure 12. The surface shape remains roughly the same after polishing for 4.6 min in the polishing range −4.2 mm to 4.2 mm, and the average material removal depth is $0.901\times 10^{-3}\;\mathrm{mm}$.

## 4. Experiment and Analysis

To begin, we polished mold one using the parameters in Table 1. The surface PV converged from 0.3572 μm to 1292 μm (Figure 13a–c) after two polishing processes of 16.6 min and 10.1 min, respectively. Meanwhile, the SR distribution varied with radius after polishing with constant rotation speed mode, which produced scattering and reflectivity at different angles on the lenses, and seriously affected the optical performance [14]. It is possible that the workpiece axis linear velocity is larger at the edge of the mold, which leads to an increase in rotational instability, resulting in the fluctuation of the polishing tool TIF at the edge. On the other hand, the material removal rate per unit time is larger due to the increasing linear velocity at the edge. Therefore, the surface error fluctuation value will become worse after the convolution iteration, which will eventually lead to an uneven state of SR distribution with the change of the workpiece axis velocity. Through analysis, it can be seen that the workpiece axis speed is a process parameter that changes constantly during the constant rotation speed mode. First, we measured and analyzed the SR of the plane mold sample of the ring band experiment in Table 1 to verify the influence of the factor on the SR of tungsten carbide mold (The ring band experiment in Table 1 is an exploration using the workpiece axis speed as a single variable). Second, we took a sampling point for the center and left and right waist of each U-shaped ring band in Figure 10a, and obtained the mean SR of each ring band. However, the SR of the ring band is relatively large when the workpiece axis speed is 10 mm/s, as shown in Figure 14, which may be the reason that the first ring band is not polished enough to contact the diamond particles. Therefore, this results in the Ra distribution being closer to the original SR of the plane mold.
(22)$$C_{V}=\frac{S_{D}}{M_{N}}\times 100\%.$$

$C_{V}$ is the SR dispersion coefficient, $S_{D}$ stands for the standard deviation of SR, and$\;M_{N}$ refers to the mean value of SR.

**Figure 13 materials-15-04848-f013:**
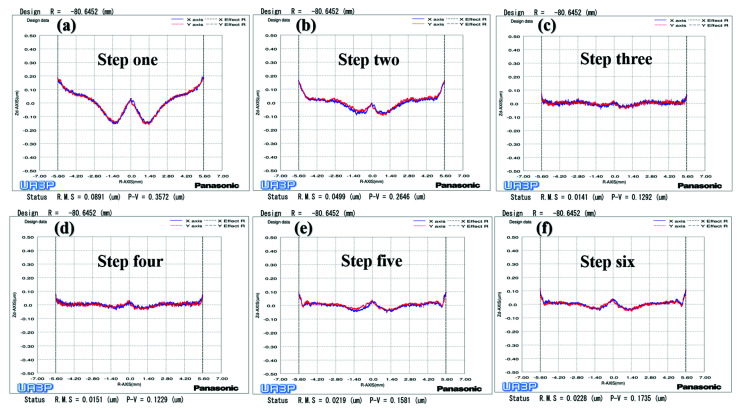
Surface shape change process of aspherical mold one: (**a**) Grinding; (**b**) The first polishing with constant rotation speed mode; (**c**) The second polishing with constant rotation speed mode; (**d**) Polishing with variable speed mode; (**e**) Polishing with small polishing disc tool; (**f**) Polishing with variable speed mode.

**Figure 14 materials-15-04848-f014:**
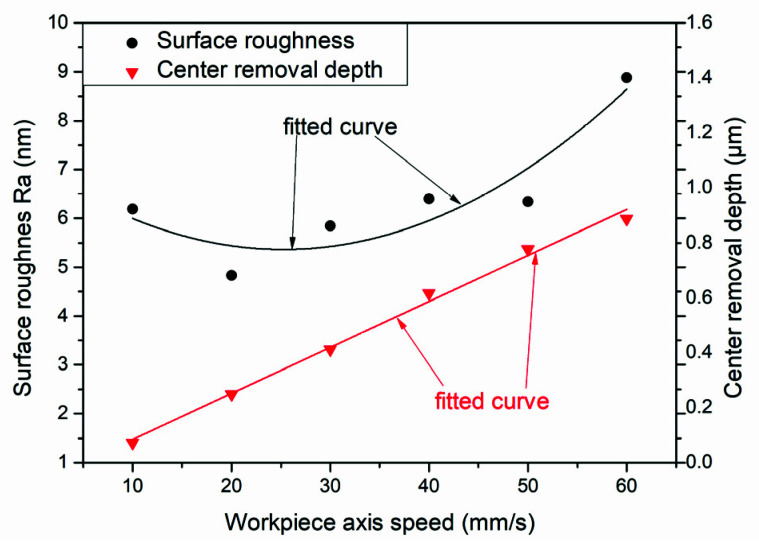
The relationship between SR and center removal depth with workpiece axis speed.

Through the above analysis, it can be concluded that the removal depth and SR will increase as the workpiece axis speed increases. Therefore, we uniformly polished the mold of the previous step for 9 min through the fourth step variable speed polishing mode, while the surface shape remained unchanged (Figure 13d). Next, we used the UA3P-300 (the measurement parameters in Table 1) to measure the SR of the aspherical center, middle, and edge before and after each polishing, as shown in the positions in Figure 15a, and obtained the $M_{N}$ and $S_{D}$ of the SR. Furthermore, it can be seen that the SR distribution tends to be uniform after polishing in variable speed mode (Figure 15a,b). However, it was found that the small spherical polishing tool was not effective for reducing SR. This is because the ring band TIF of the spherical polishing tool is relatively small and conforms to the Gaussian distribution, which is conducive to the correction of the surface shape of the mold. However, there are tiny waves after convolution and iteration of its TIF, combined with the polishing trajectory of its screw feed, as shown in the Figure 5d, which shows the performance of the HFE of the mold. Then in step five, the small polishing disc tool bonded with diamond particles was used for uniform polishing with the process parameters in Table 1. Since mold one is an m-shaped aspherical shape with micro inversion structure, as shown in the 3D image of Figure 5c, interference occurs when the small polishing tool moves to the edge. Thus, small pits and warping points appear at the edge of the surface (Figure 13e). The SR of the edge interference site is large (Figure 15c), which also increases the $C_{V}$ after polishing in this step (Figure 16). However, it can also be seen that the advantages of the smoothing effect of the small polishing disc tool lie in its quick reduction of the SR, enabling most regions of the aspherical surface to be reduced to less than 2 nm after 3.4 min of uniform polishing. Finally, we polished the mold in the previous step for 9 min with a small spherical polishing tool in variable speed mode in the sixth step. The surface shape can remain roughly the same (Figure 13f), and the warped edge SR converged (Figure 15d). Moreover, the overall SR tended to increase slightly, but the uniformity is improved, and the $C_{V}$ decreased from 27.9% to 14.2% (Figure 16). Through the experiment, the following three conclusions can be drawn: first, the constant rotation speed polishing mode of the small spherical polishing tool has a good effect on the profile shape correction, but easily leads to uneven SR distribution. Second, the variable speed polishing mode of the small spherical polishing tool is beneficial in improving the overall SR uniformity. Third, the smoothing polishing mode of the small polishing disc tool can roughly maintain the mold profile shape and quickly reduce the SR.

To verify the above conclusions, we further tested the aspherical mold two in Figure 2a. The experiment results are shown in Figure 17, Figure 18 and Figure 19, which describe the change process of surface shapes, explain the SR distribution, and illustrate the change process of SR and its $C_{V}$. In step one, we obtained an initial mold with an uneven SR distribution after grinding (Figure 18a). In step two, the surface PV was roughly maintained while both the overall SR and the $C_{V}$ were reduced after polishing with a small polishing disc tool for two minutes. However, the SR was still had a small center and a large edge (Figure 17a,b, Figure 18b, and Figure 19). Next, in step three, the surface PV was roughly maintained after uniform polishing with the constant rotation speed mode for 5 min (Figure 17b,c), but both the SR and the $C_{V}$ tended to increase (Figure 18c and Figure 19). This phenomenon further proves that it is easy to cause uneven SR distribution with that polishing mode. Finally, in step four, the surface PV converged from 0.2075 μm to 0.071 μm after polishing for 17.5 min with the variable speed mode (Figure 17c,d). The uniformity of the overall SR improved and the $C_{V}$ was reduced from 41.6% to 12.7% (Figure 18d and Figure 19). However, it was found that the surface PV correction efficiency of the variable speed mode was low, and the polishing time was 3.1 times that of the constant rotation speed mode with the same process parameters from the simulation (take Figure 18c as an example).

## 5. Conclusions

In conclusion, the paper aims at the correction of the surface PV and the improvement of the SR distribution of the small aspherical mold. First, a small spherical polishing tool and a semi-flexible disc polishing tool were designed, and their TIF was established. Then, a constant speed and variable speed combined polishing method for screw feed was proposed. Two aspherical molds with an optical effective diameter less than 13 mm were experimentally verified on a self-developed polishing machine. The results show that the constant rotational speed polishing mode has a faster shaping efficiency, but the linear velocity of the workpiece axis changes with the polishing position in real time, resulting in uneven roughness distribution. The variable speed polishing mode ensures that the axis speed of the workpiece at most positions is constant, which is beneficial to the improvement of the overall roughness uniformity. The disc polishing tool with microstructure has a smoothing effect, which can quickly reduce the roughness of the aspherical surface. Finally, the surface PVs of the two molds were 0.3572 μm and 0.2075 μm, converging to 0.1282 μm and 0.071 μm, respectively. Meanwhile, the $C_{V}$ decreased from 27.9% and 41.6% to 14.2% and 12.7% respectively.

The study provides a better theoretical and practical machining strategy for high-precision polishing of small aspherical molds, especially for research on shape correction and SR distribution uniformity improvement. However, the surface repair of small spherical polishing tools usually requires multiple compensations to converge to the target PV.

Future investigations will be focused on the TIF accuracy of small spherical polishing tools. It is hoped that a nonlinear TIF with time and aspherical curvature changes will be proposed to meet the polishing of aspherical molds with different shapes and large curvature changes, and to improve the polishing efficiency.

## Figures and Tables

**Figure 1 materials-15-04848-f001:**
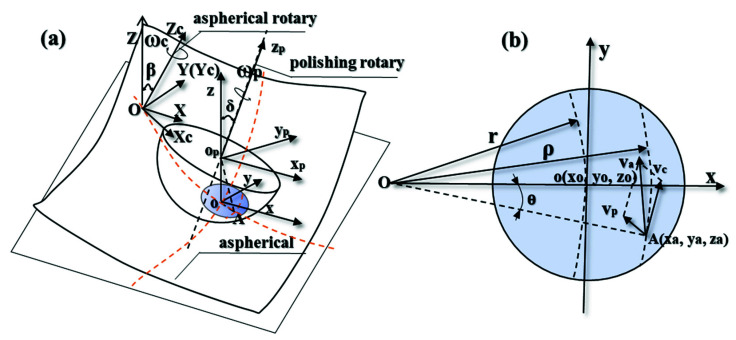
(**a**) Schematic diagram of spherical polishing tool; (**b**) Schematic diagram of the relative line speed distribution in the contact region.

**Figure 2 materials-15-04848-f002:**
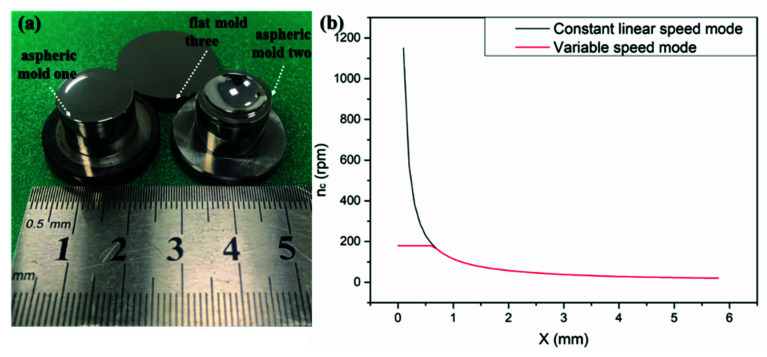
(**a**) Experimental mold; (**b**) Changes of the workpiece axis speed in the variable speed mode with the radius of the aspherical mold.

**Figure 3 materials-15-04848-f003:**
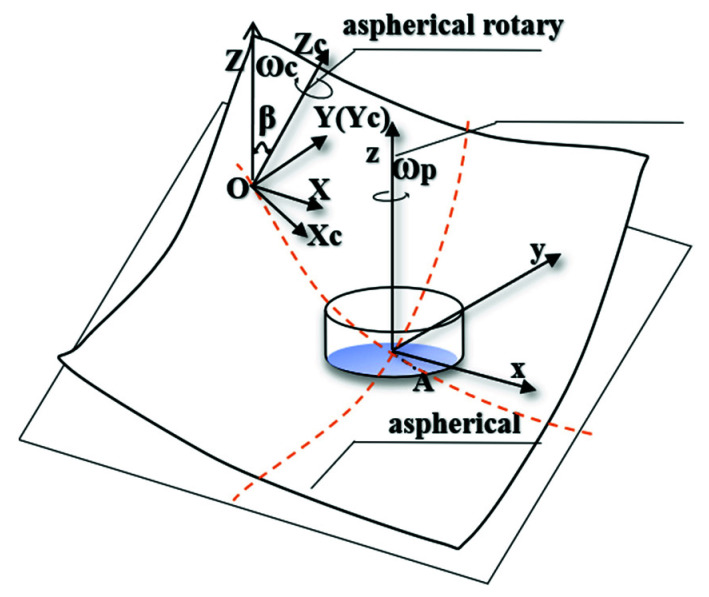
Schematic polishing of small polishing disc tool.

**Figure 4 materials-15-04848-f004:**
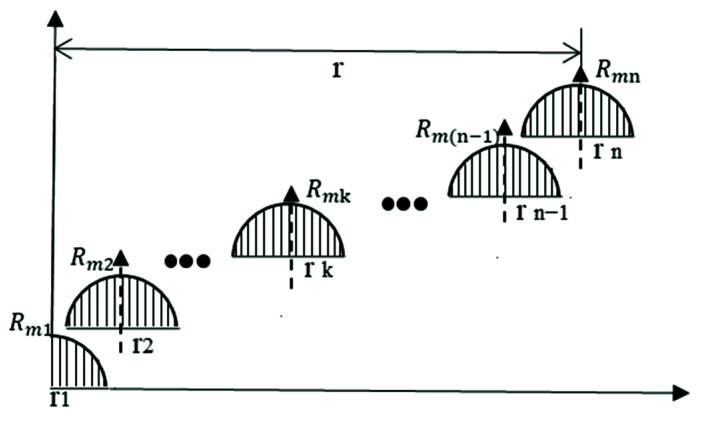
Sketch of the superposition of the discrete TIF at each sample point.

**Figure 5 materials-15-04848-f005:**
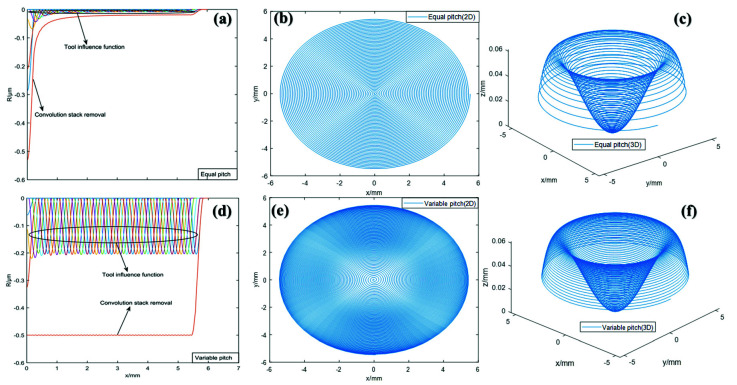
Simulation diagram of the superposition of the discrete TIF and screw feed polishing trajectory: (**a**) Equal pitch TIF; (**b**) Equal pitch polishing trajectory (2D); (**c**) Equal pitch polishing trajectory (3D); (**d**) Variable pitch TIF; (**e**) Variable pitch polishing trajectory (2D); (**f**) Variable pitch polishing trajectory (3D).

**Figure 6 materials-15-04848-f006:**
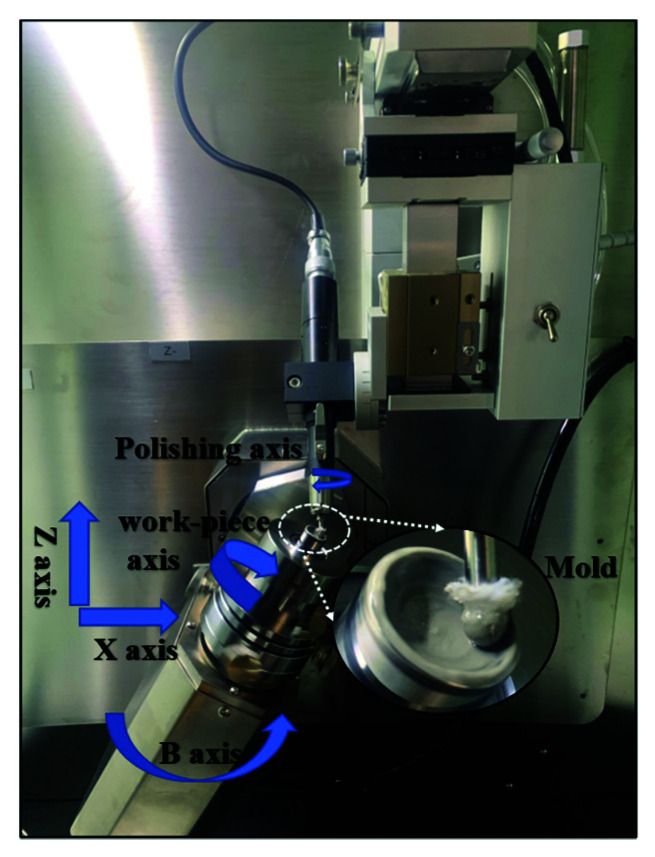
Self-developed polishing equipment.

**Figure 7 materials-15-04848-f007:**
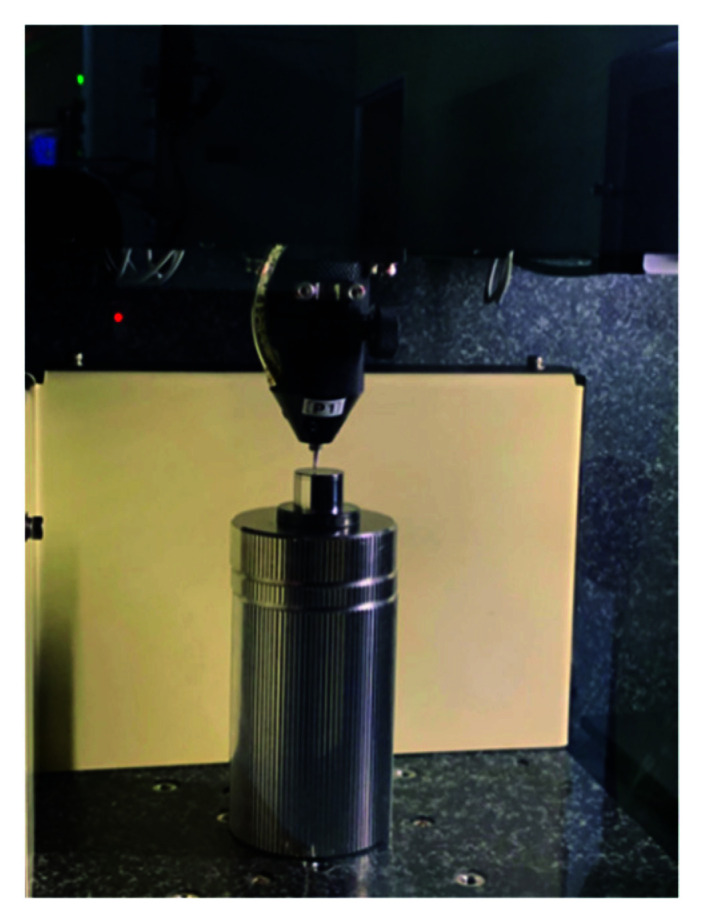
Mold measurement.

**Figure 8 materials-15-04848-f008:**
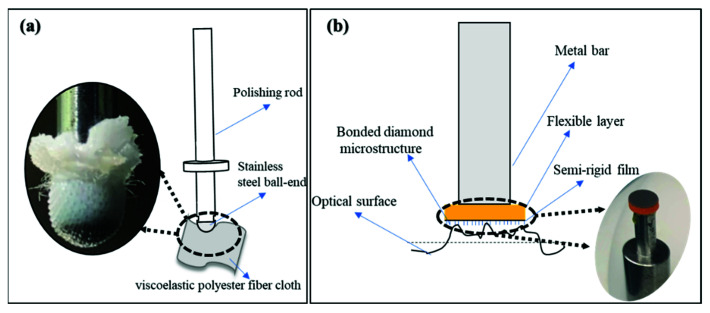
(**a**) Small spherical polishing tool structure; (**b**) Semi-flexible small polishing disc tool structure.

**Figure 9 materials-15-04848-f009:**
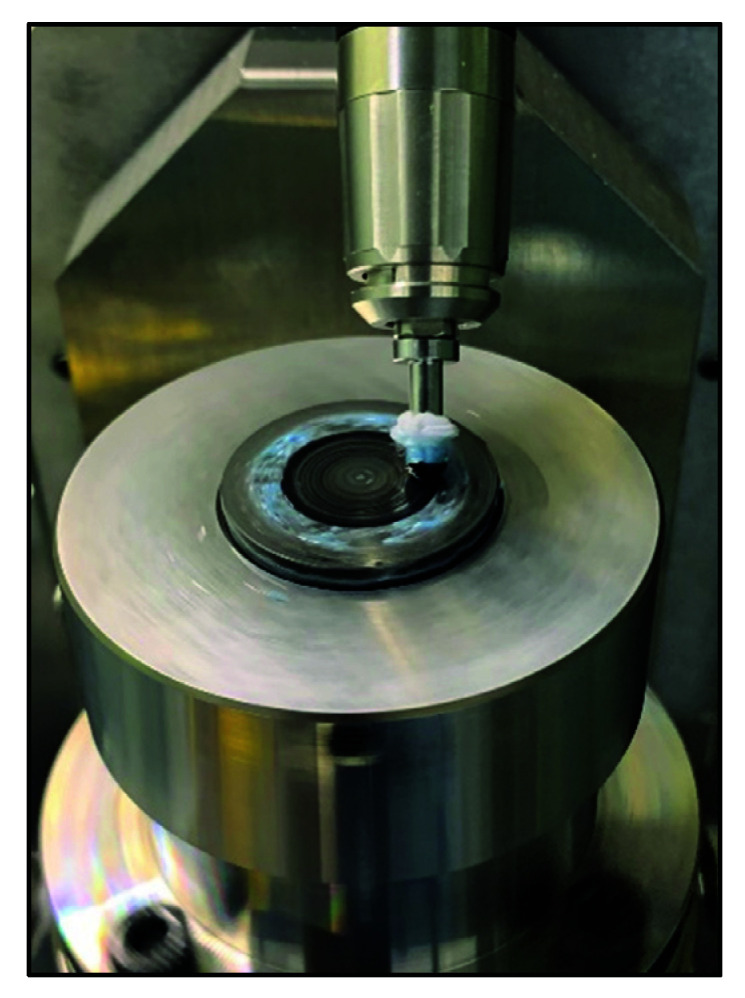
Ring experiment of spherical polishing tool.

**Figure 10 materials-15-04848-f010:**
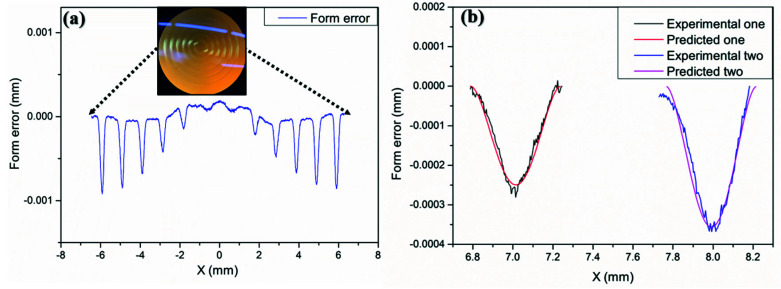
(**a**) Experimental contour survey of ring band; (**b**) Profile shape comparison of experimental and theoretical model.

**Figure 11 materials-15-04848-f011:**
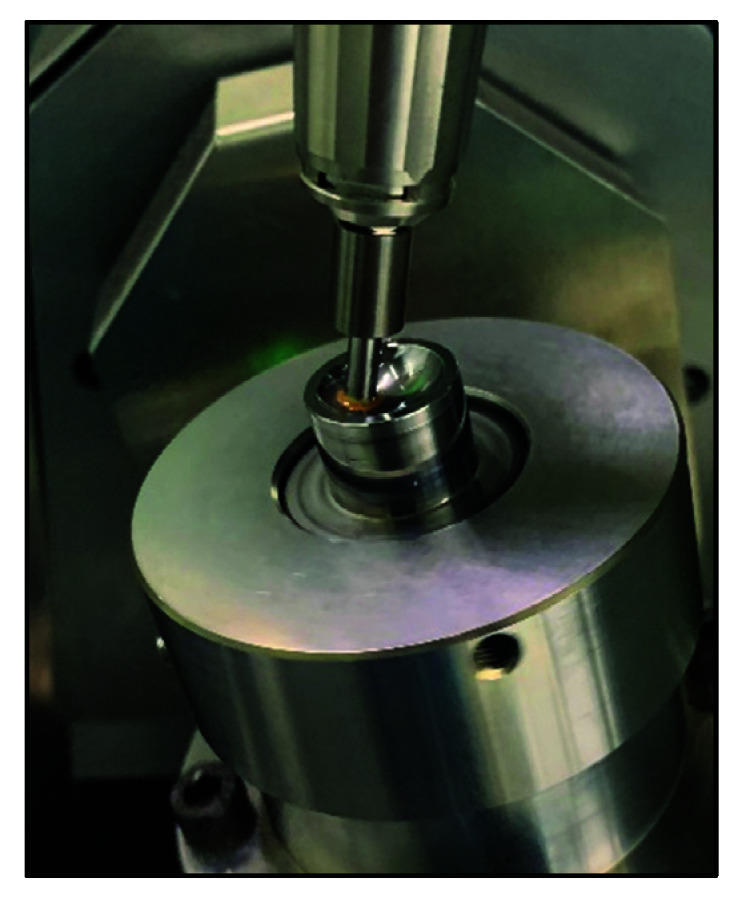
Uniform polishing experiment of disc polishing tool.

**Figure 12 materials-15-04848-f012:**
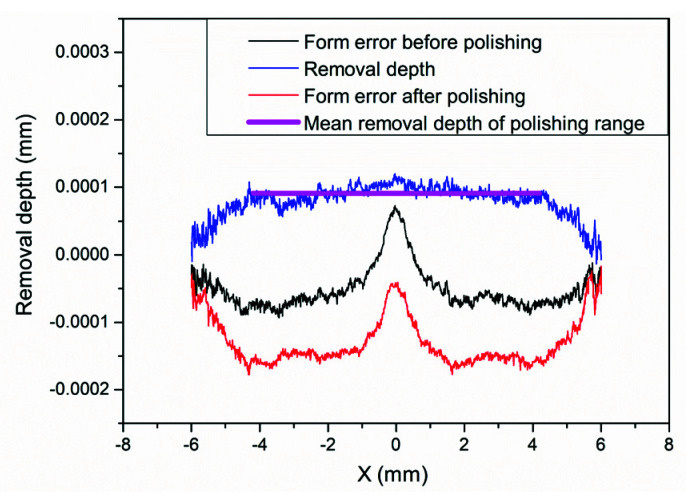
Profile comparison of aspherical mold for uniform polishing with small polishing disc tool.

**Figure 15 materials-15-04848-f015:**
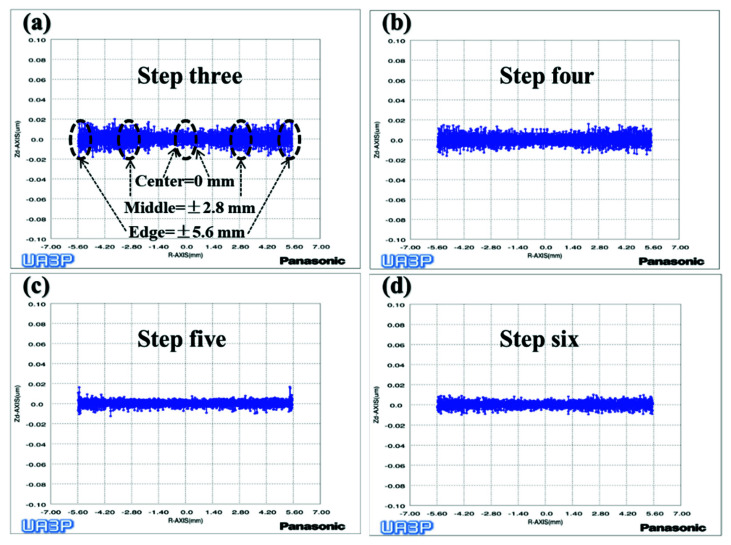
Process of SR distribution change of aspherical mold one: (**a**) Polishing with constant rotation speed mode; (**b**) Polishing with variable speed mode; (**c**) Polishing with small polishing disc tool; (**d**) Polishing with variable speed mode.

**Figure 16 materials-15-04848-f016:**
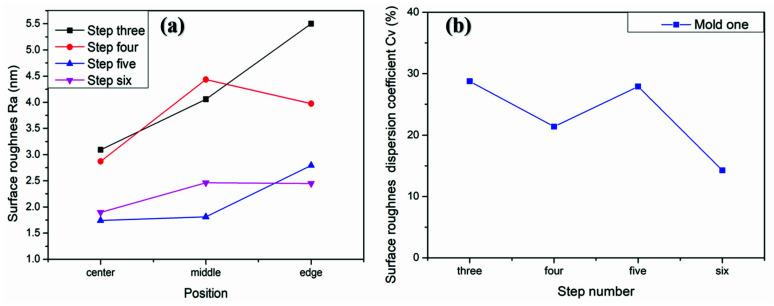
Mold one: (**a**) Comparison diagram of SR changes in the center, middle and edge of aspherical surface during polishing process; (**b**) Change diagram of SR dispersion coefficient.

**Figure 17 materials-15-04848-f017:**
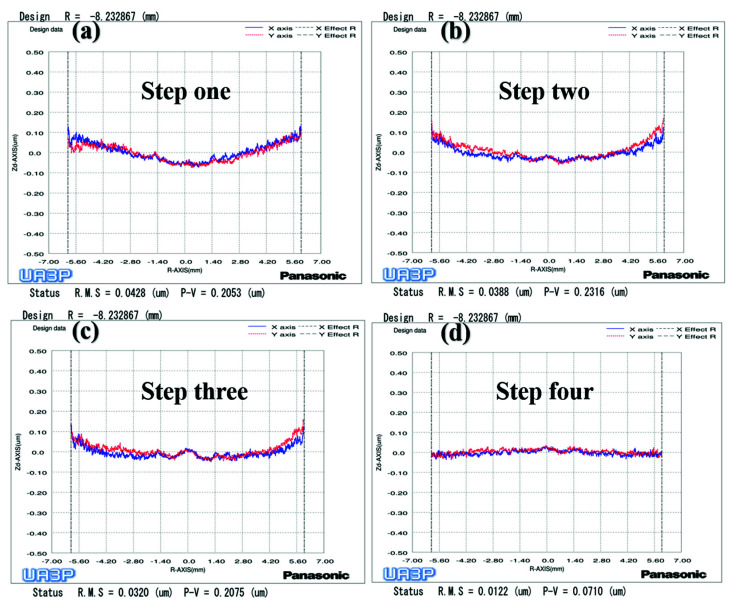
Surface shape change process of aspherical mold two: (**a**) Grinding (**b**) Polishing with small polishing disc tool; (**c**) Polishing with constant rotation speed mode; (**d**) Polishing with variable speed mode.

**Figure 18 materials-15-04848-f018:**
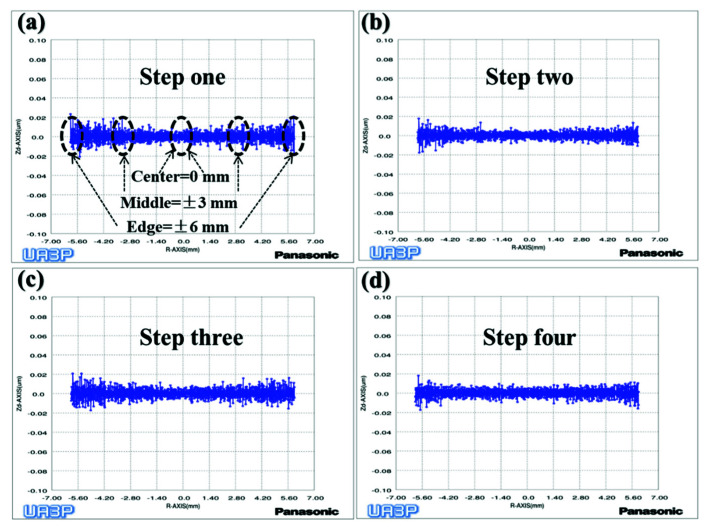
Process of SR distribution change of aspherical mold two: (**a**) Grinding (**b**) Polishing with small polishing disc tool; (**c**) Polishing with constant rotation speed mode; (**d**) Polishing with variable speed mode.

**Figure 19 materials-15-04848-f019:**
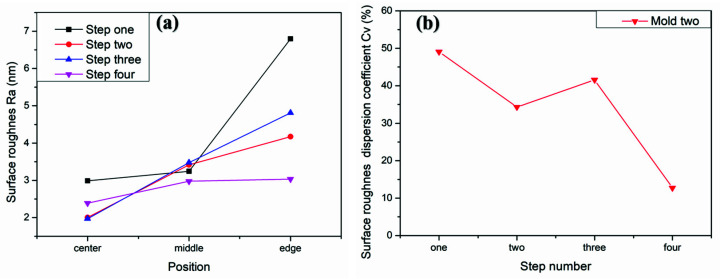
Mold two: (**a**) Comparison diagram of SR changes in the center, middle, and edge of the aspherical surface during polishing process; (**b**) Change diagram of SR dispersion coefficient.

**Table 1 materials-15-04848-t001:** Polishing and test conditions.

Item	Condition
Workpiece	Material: Tungsten Carbide, $E_{2}=615\;\mathrm{GPa}$ $,\;\varepsilon_{2}=0.3$
Slurry	Material: diamond compound, Brand: Hyprez, particle size =1 μm
Disc polishing tool	Structure: Semi-flexible microstructure of bonded diamond particles, bonded diamond size =1 μm $,\;r_{p}=2\;\mathrm{mm}$ $,\;n_{p}=250\;\mathrm{rpm}$ $,\;n_{c}=54\;\mathrm{rpm}$ , F=0.18 N, σ=0 °
Spheric polishing tool	Material: viscoelastic polyester fiber cloth wrapped steel ball, $\;E_{1}=39.4\;\mathrm{GPa}$ $,\;\varepsilon_{1}=0.32$ $,\;r_{p}=1.5\;\mathrm{mm}$ $,\;n_{p}=200\;\mathrm{rpm}$ $,\;n_{c}=180\;\mathrm{rpm}$ , F=0.18 N, σ=35 °
UA3P-300 SR test	Roughness analysis cutoff: Wc = 80 μm, range (length) selected for SR analysis: L/Wc = 5, Wc cutoff sample number = 40 pts

**Table 2 materials-15-04848-t002:** Parameters and results of the ring band experiment.

Ring Band Polishing Radius *r* (mm)	Polishing Time *T* (s)	Workpiece Line Speed $\;v_{c}\;$(mm/s)	Removal Coefficient *k* $\left({\mathrm{mm}}^{2}/N\right)$
1	37.7	10	$1.251\times 10^{-6}$
2	75.4	20	$3.491\times 10^{-6}$
3	113.0	30	$4.62\times 10^{-6}$
4	150.8	40	$5.633\times 10^{-6}$
5	188.5	50	$5.94\times 10^{-6}$
6	226.2	60	$5.802\times 10^{-6}$

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
