# Peer review of "Surface Quality Control Strategy of Aspherical Mold Based on Screw Feed Polishing of Small Polishing Tool"

_materials, 2022, doi:10.3390/ma15144848_

Round 1
Reviewer 1 Report
Dear authors, the manuscript ‘Surface quality control strategy of aspherical mold based on screw feed polishing of small polishing tool’, Manuscript ID: materials-1788538, have some weakness that must(!) be improved significantly.
Please find some of the most crucial suggestions, as listed below:
1. From the ‘Introduction’ part, even though the proposal(s) was defined, it is difficult to conclude the lack of presented studies in the critical review of the current state of knowledge, if exists. From that proposal, the novelty seems to be lost, even properly defined, as mentioned previously.
2. There is no word where the equations (1)-(17) were taken from? If published previously must be cited but, respectively, if proposed by the author(s), it should(!) be indicated as well. From that matter, the whole section 2 can be classified as a novelty (or not?). In some sentences, like ‘From Figure 2, it can be seen that the rotational speed and the growth speed of the aspherical central area are relatively large, so after comprehensively considering the acceleration and maximum speed of the polishing equipment, set the maximum speed of the work-piece axis as ?? = 180 ???. Therefore, this study explores the variable speed mode made up by the constant line speed and the constant rotation speed of the contact region.’, page 4, does not clearly define this matter.
3. Some results and the primary conclusion are presented in Figure 2. It is not clear if those results are general, received e.g. from other (already published) studies or found by the author(s). It must be resolved and properly presented.
4. Even though the equation (21) derives from some sentences, its application was not clearly justified. Maybe sentence ‘Next, we measure the surface roughness of the aspherical center, middle and edge before and after each polishing, and obtain the surface roughness dispersion coefficient corresponding to step three to step six according to equation (21)’?
5. There is no word about the surface roughness (SR) measurement process but only ‘Next, we measure the surface roughness of the aspherical center, middle and edge before and after each polishing, and obtain the surface roughness…’. Moreover, any details against SR measurement uncertainty (1) or measurement (2) or high-frequency (3) noise were not addressed. Please look for some examples:
(1) https://doi.org/10.1088/2051-672X/3/3/035004
(2) https://doi.org/10.1088/0957-0233/23/3/035008
(3) https://doi.org/10.3390/ma14175096
6. The ‘Conclusion’ section is interesting and, correspondingly, provides required and valuable information, there are no further proposals, even assumptions that would significantly confirm the meaning of the studies presented. Please try to emphasize the most valuable future prospects.
Moreover, some editorial issues should be even mentioned:
7. On page 2 there is ‘making= it reasonable’ instead of ‘making it reasonable’, I suppose (?).
8. There is a double space on page 10 in the sentence ‘Secondly, we take a…’.
9. Even the template requirements of the Materials journal were not provided at this stage of the review process, the DOI links should be included in the references.
Generally, the proposed manuscript has some weaknesses and, at least in the current form, is not suitable for publication in a quality journal as the Materials is, it must be improved significantly (!) before any further processing, if allowed.
Author Response
Dear Reviewers:
Thanks for your comments concerning our manuscript entitled “Machining strategy of aspherical mold based on screw feed polishing of small polishing tool” (ID: materials- 1788538).
Those comments are valuable and very helpful for improving our paper. We have studied comments carefully and made corrections which we hope meet your acquirements. Revised parts are marked in red in the revised manuscript. The main corrections in the paper and the responses to the reviewer’s comments are listed as follows:
Comment 1: From the ‘Introduction’ part, even though the proposal(s) was defined, it is difficult to conclude the lack of presented studies in the critical review of the current state of knowledge, if exists. From that proposal, the novelty seems to be lost, even properly defined, as mentioned previously.
Response: The introduction has been reedited, including the relevant research progress and limitations, and puts forward the necessity of this work.
Comment 2: There is no word where the equations (1)-(17) were taken from? If published previously must be cited but, respectively, if proposed by the author(s), it should(!) be indicated as well. From that matter, the whole section 2 can be classified as a novelty (or not?). In some sentences, like ‘From Figure 2, it can be seen that the rotational speed and the growth speed of the aspherical central area are relatively large, so after comprehensively considering the acceleration and maximum speed of the polishing equipment, set the maximum speed of the work-piece axis as ?? = 180 ???. Therefore, this study explores the variable speed mode made up by the constant line speed and the constant rotation speed of the contact region.’, page 4, does not clearly define this matter.
Response: We have added references to Equation 1 and Equation 2. The other equations are derived and built based on our polish way, location and trajectory. Then, we add descriptions and definitions regarding the variable speed mode, as in line 192 and Equation 12.
Comment 3: Some results and the primary conclusion are presented in Figure 2. It is not clear if those results are general, received e.g. from other (already published) studies or found by the author(s). It must be resolved and properly presented.
Response: Figure 2 is drawn by Equation 11 and Equation 12 whose description in the paper has been modified in lines 189 and 194.
Comment 4: Even though the equation (21) derives from some sentences, its application was not clearly justified. Maybe sentence ‘Next, we measure the surface roughness of the aspherical center, middle and edge before and after each polishing, and obtain the surface roughness dispersion coefficient corresponding to step three to step six according to equation (21)’?
Response: We have supplemented the description and definition of this equation in line 355-357.
Comment 5: There is no word about the surface roughness (SR) measurement process but only ‘Next, we measure the surface roughness of the aspherical center, middle and edge before and after each polishing, and obtain the surface roughness…’. Moreover, any details against SR measurement uncertainty (1) or measurement (2) or high-frequency (3) noise were not addressed. Please look for some examples:
(1) https://doi.org/10.1088/2051-672X/3/3/035004
(2) https://doi.org/10.1088/0957-0233/23/3/035008
(3) https://doi.org/10.3390/ma14175096
Response: We improve the details of the measurement, and add the measurement parameters for the SR Ra measured by the UA3P-300 in Table 1. Figures 15 and 18 can intuitively observe the global distribution of roughness, and then we also marked the measurement positions corresponding to the center, middle and edge in the figure. In this paper, the experiments are carried out on two molds, also to eliminate the uncertainty of the roughness measurement of the equipment.
In addition, because this is a scientific research project in cooperation with the enterprise, who requires the analysis results of Panasonic's UA3P as the indicator. The SR measurement and analysis in this paper are directly measured by the parameters in Table 1.
Comment 6: The ‘Conclusion’ section is interesting and, correspondingly, provides required and valuable information, there are no further proposals, even assumptions that would significantly confirm the meaning of the studies presented. Please try to emphasize the most valuable future prospects. Moreover, some editorial issues should be even mentioned:
Response: The conclusions have been rewritten. The major findings are summarized and future prospects are put forward. The valuable contribution is that we provide a better theoretical and practical machining strategy for high-precision polishing of small aspheric mold, especially in the research of shape correction and SR distribution uniformity improvement.
Comment 7: On page 2 there is ‘making= it reasonable’ instead of ‘making it reasonable’, I suppose (?).
Response: This mistake has been corrected in line 109. Some grammatical errors and typo-mistakes have been corrected, too.
Comment 8: There is a double space on page 10 in the sentence ‘Secondly, we take a…’.
Response: This mistake has been corrected.
Comment 9: Even the template requirements of the Materials journal were not provided at this stage of the review process, the DOI links should be included in the references.
Response: We have rearranged the typesetting according to the template requirements of the journal, and added DOI links in the references.
Thank you for your detailed reviews and direction again. We will do my best to improve the manuscript.
Best regards,
Honghui Yao
Reviewer 2 Report
Reviewed article concerns surface quality control strategy of aspherical mold based on screw feed polishing of small polishing tool and is write in accordance with generally accepted standards of the scientific works. After careful reading of the submitted text there are some substantive remarks that should be taken into consideration by the Authors to improve reviewed text.
1. The abstract should include information about new methods, results, concepts, and conclusions – in its current form, the abstract needs to be rewritten to include more information about motivation of research conducted. Also, in its current form abstract did not refer to “surface quality control strategy” (given in title of the work). It is more about machining strategy than quality control. Perhaps the title of the work should be modified to better reflect its content.
2. None of the key words refer to quality control given as the main term in the title of the work.
3. The novelty of given approach should be emphasized in introduction. Also, the scientific problem that the work addresses must be clearly stated.
4. The Authors should provide more precise information about used experimental and measurement positions. I suggest providing all main parameters and conditions of experimental test in a form of table.
5. Presented study widely covers defined scientific problem and with simulation and experimental investigations provides proper background for given conclusions, however deeper scientific consideration of obtained results referred to the basic phenomena in polishing processes should be given.
6. I suggest also to give wider description of potential use of presented findings in scientific research as well as in industrial practice.
7. The strengths and limitations of the obtained results and applied methods should be clearly described.
8. In conclusions deeper explanation of observed phenomena should be given (conclusions should refer not only to results but also to causes of obtained results).
9. The conclusions should highlight the novelty and contribution to the state of the knowledge in given area.
After a careful study of the text sent for review, many editorial comments also come to mind:
‒ text of scientific rapports should be writing impersonal,
‒ synthetic list of main nomenclature (symbols and acronyms) would improve readability of this study,
‒ all mathematical/physical symbols should be writing italics and proper subscript/superscript notation for better readability of the text,
‒ figures should be given in the text directly after first mentioning, not half page before like Fig. 1,
‒ some of the figures were not mentioned in the text at all – if a figure does not contribute to the text, it should not be included in the work,
‒ correct the numbering of tables in the references in the text,
‒ the names of roughness parameters and their symbols should be used consistently (e.g., the undefined “roughness” parameter in Figure 16),
‒ illegible axis descriptions on some figures (e.g., Fig. 14),
‒ lack of punctuation marks after equitation (equitation is part of a sentence).
Author Response
Dear Reviewers:
Thanks for your comments concerning our manuscript entitled “Machining strategy of aspherical mold based on screw feed polishing of small polishing tool” (ID: materials- 1788538).
Those comments are valuable and very helpful for improving our paper. We have studied comments carefully and made corrections which we hope meet your acquirements. Revised parts are marked in red in the revised manuscript. The main corrections in the paper and the responses to the reviewer’s comments are listed as follows:
Comment 1: The abstract should include information about new methods, results, concepts, and conclusions – in its current form, the abstract needs to be rewritten to include more information about motivation of research conducted. Also, in its current form abstract did not refer to “surface quality control strategy” (given in title of the work). It is more about machining strategy than quality control. Perhaps the title of the work should be modified to better reflect its content.
Response: The abstract has been reedited, and the title has been modified. We initially wanted to define the surface PV and surface roughness Ra of the mold as surface quality control. But it is indeed more appropriate to change the surface quality control to a machining strategy after your suggestion.
Comment 2: None of the key words refer to quality control given as the main term in the title of the work.
Response: The title and keywords have been revised.
Comment 3: The novelty of given approach should be emphasized in introduction. Also, the scientific problem that the work addresses must be clearly stated.
Response: Yes, the introduction has also been improved, including the relevant research progress and limitations, and puts forward the necessity of this work.
Comment 4: The Authors should provide more precise information about used experimental and measurement positions. I suggest providing all main parameters and conditions of experimental test in a form of table.
Response: All of the experimental and measurement detailed parameters have been improved in Table 1, Table 2, Table 3 and Table 4. The specific measurement position of Ra has been marked in the Fig. 15 (a) and Fig. 18 (a).
Comment 5: Presented study widely covers defined scientific problem and with simulation and experimental investigations provides proper background for given conclusions, however deeper scientific consideration of obtained results referred to the basic phenomena in polishing processes should be given.
Response: Yes, we have taken a deeper scientific consideration of the main results obtained on the basic phenomena in polishing processes. For example, in lines 325 and 360, etc.
Comment 6: I suggest also to give wider description of potential use of presented findings in scientific research as well as in industrial practice.
Response: The potential use of presented findings in scientific research as well as in industrial practice has been re-emphasized in the abstract, introduction and conclusion. In general, this paper mainly solves the problems of difficult surface PV repair and poor SR and SR distribution uniformity of small-diameter aspheric molds and provides a good machining solution for high-quality precision polishing of small-diameter aspheric molds.
Comment 7: The strengths and limitations of the obtained results and applied methods should be clearly described.
Response: Line 376 depicts the strengths and limitations of the individual polishing modes. The paper combines the advantages of each to take into account the processing quality and efficiency of small-diameter aspheric molds at the same time. Finally, it is mentioned in the conclusion that the accuracy of the surface correction of the spherical small polishing tool is not enough, and in-depth research will be carried out in the future.
Comment 8: In conclusions deeper explanation of observed phenomena should be given (conclusions should refer not only to results but also to causes of obtained results).
Response: The conclusions have been rewritten. We put the reason analysis corresponding to the result in the text of the experiment and analysis part, taking into account the length of the conclusion.
Comment 9: The conclusions should highlight the novelty and contribution to the state of the knowledge in given area.
Response: The conclusions have been rewritten. The main contribution is that we provide a better theoretical and practical machining strategy for high-precision polishing of small aspheric mold, especially in the research of shape correction and SR distribution uniformity improvement.
Comment 10: After a careful study of the text sent for review, many editorial comments also come to mind:
‒ text of scientific rapports should be writing impersonal,
‒ synthetic list of main nomenclature (symbols and acronyms) would improve readability of this study,
‒ all mathematical/physical symbols should be writing italics and proper subscript/superscript notation for better readability of the text,
‒ figures should be given in the text directly after first mentioning, not half page before like Fig. 1,
‒ some of the figures were not mentioned in the text at all – if a figure does not contribute to the text, it should not be included in the work,
‒ correct the numbering of tables in the references in the text,
‒the names of roughness parameters and their symbols should be used consistently (e.g., the undefined “roughness” parameter in Figure 16),
‒ illegible axis descriptions on some figures (e.g., Fig. 14),
‒ lack of punctuation marks after equitation (equitation is part of a sentence).
Response: We have made targeted revisions one-to-one, regarding these fundamental editorial errors raised by the reviewers and combined with the editorial comments of the other three experts, such as the part marked in red in the manuscript.
Thank you for your detailed reviews and direction again. We will do my best to improve the manuscript.
Best regards,
Honghui Yao

Reviewer 3 Report
The authors presented an article “Surface quality control strategy of aspherical mold based on screw feed polishing of small polishing tool”. The article can be interesting from an engineering point of view, however there are many points in the article that require further explanation. First of all the article is not written in accordance with the guidelines for authors, therefore it is difficult, for example, to indicate the line of the text to be discussed. The article contains many technical and editorial errors without correction, the article is not suitable for publication in Materials. The article must be additionally proofread by a native English speaker.
Author Response
Dear Reviewers:
Thanks for your comments concerning our manuscript entitled “Machining strategy of aspherical mold based on screw feed polishing of small polishing tool” (ID: materials- 1788538).
The review expert gave the opinion of rejecting the manuscript. However, the comment is valuable and very helpful for improving our paper. We have studied comments carefully and made corrections which we hope meet your acquirements. Revised parts are marked in red in the revised manuscript. The main corrections in the paper and the responses to the expert’s comments are listed as follows:
Comment: The authors presented an article “Surface quality control strategy of aspherical mold based on screw feed polishing of small polishing tool”. The article can be interesting from an engineering point of view, however there are many points in the article that require further explanation. First of all the article is not written in accordance with the guidelines for authors, therefore it is difficult, for example, to indicate the line of the text to be discussed. The article contains many technical and editorial errors without correction, the article is not suitable for publication in Materials. The article must be additionally proofread by a native English speaker.
Response: Thanks to the expert comment, the manuscript has been rewritten in accordance with the author's guidelines. Then the abstract, introduction, conclusion and some editorial errors are revised based on the opinions of the other three review experts. Then, we have made our effort to enhance academic writing.
Thank you for your comments and guidance. We will do my best to improve the manuscript.
Best regards,
Honghui Yao
Reviewer 4 Report
Manuscript ID: materials-1788538
The current manuscript looks interesting. The reported results are reliable and seem to be correct. The reported results have been well described and interpreted. Thus, I recommend the paper for publication after the authors perform some minor points:
1) The paper contains some grammatical errors and typo-mistakes that should be corrected. The English language should be improved.
2) The Abstract part should be improved. It should clearly inform the important findings in the present study. The abstract should contain some qualitative and quantitative results.
3) The introduction part can be further improved. In the introduction part, the authors are suggested to include recent references by reporting the main findings in comparable studies and what is achieved until today.
4) Check the symbols and equations.
5) Could the authors provide some experimental topographical images?
6) Shorten the conclusion part. Summarize the major findings and future perspectives of the present work in the conclusion section.
Author Response
Dear Reviewers:
Thanks for your comments concerning our manuscript entitled “Machining strategy of aspherical mold based on screw feed polishing of small polishing tool” (ID: materials- 1788538).
Those comments are valuable and very helpful for improving our paper. We have studied comments carefully and made corrections which we hope meet your acquirements. Revised parts are marked in red in the revised manuscript. The main corrections in the paper and the responses to the expert’s comments are listed as follows:
Comment 1: The paper contains some grammatical errors and typo-mistakes that should be corrected. The English language should be improved.
Response: Some basic grammatical errors as well as the typo-mistakes have been revised, then we have made our effort to enhance academic writing.
Comment 2: The Abstract part should be improved. It should clearly inform the important findings in the present study. The abstract should contain some qualitative and quantitative results.
Response: Yes, the abstract has been rewritten.
Comment 3: The introduction part can be further improved. In the introduction part, the authors are suggested to include recent references by reporting the main findings in comparable studies and what is achieved until today.
Response: Yes, the introduction has also been improved, including the relevant research progress and limitations, and puts forward the necessity of this work.
Comment 4: Check the symbols and equations.
Response: Symbols and equations have been rechecked and are all italicized. Some other basic errors in the manuscript have also been corrected.
Comment 5: Could the authors provide some experimental topographical images?
Response: Yes, for example, picture 14, picture 15, picture 17 and picture 18 are experimental topographical images.
Comment 6: Shorten the conclusion part. Summarize the major findings and future perspectives of the present work in the conclusion section.
Response: The conclusion part has been shortened, and the major findings are summarized and future prospects are put forward.
Thank you for your detailed reviews and direction again. We will do my best to improve the manuscript.
Best regards,
Honghui Yao
Round 2
Reviewer 1 Report
Dear authors, manuscript ‘Surface quality control strategy of aspherical mold based on screw feed polishing of small polishing tool’, Manuscript ID: materials-1788538, has been improved in a required manner so, correspondingly, can be further processed by the Materials journal.
Thank you for your full responses that, in their current form, were addressed properly and make the manuscript more suitable for publication in a quality journal as the Materials is.
Reviewer 2 Report
All my comments were taken into consiferation.
Article can be published in its current form.